# A Virtual Multi-Ocular 3D Reconstruction System Using a Galvanometer Scanner and a Camera

**DOI:** 10.3390/s23073499

**Published:** 2023-03-27

**Authors:** Zidong Han, Liyan Zhang

**Affiliations:** College of Mechanical & Electrical Engineering, Nanjing University of Aeronautics and Astronautics, Nanjing 210016, China

**Keywords:** galvanometer, virtual camera, multi-ocular, calibration, 3D reconstruction

## Abstract

A novel visual 3D reconstruction system, composed of a two-axis galvanometer scanner, a camera with a lens, and a set of control units, is introduced in this paper. By changing the mirror angles of the galvanometer scanner fixed in front of the camera, the boresight of the camera can be quickly adjusted. With the variable boresight, the camera can serve as a virtual multi-ocular system (VMOS), which captures the object at different perspectives. The working mechanism with a definite physical meaning is presented. A simple and efficient method for calibrating the intrinsic and extrinsic parameters of the VMOS is presented. The applicability of the proposed system for 3D reconstruction is investigated. Owing to the multiple virtual poses of the camera, the VMOS can provide stronger constraints in the object pose estimation than an ordinary perspective camera does. The experimental results demonstrate that the proposed VMOS is able to achieve 3D reconstruction performance competitive with that of a conventional stereovision system with a much more concise hardware configuration.

## 1. Introduction

Vision-based three-dimensional (3D) reconstruction has been among the most popular research fields for years. This technique endeavors to recover the 3D information of spatial objects or scenes from one or multiple images. It has been widely applied in many flourishing fields, such as auto driving, industrial inspection, building modeling, etc.

Generally, the implementation of 3D reconstruction of unknown objects requires images taken from different perspectives, which may be synchronously captured by two or more spatially distributed cameras. The 3D reconstruction devices with two or more cameras are usually called binocular or multi-ocular 3D vision systems, respectively. For example, Fu et al. proposed an optimization method based on the correspondence field (CF) to determine a proper camera arrangement in a dual-camera system, which turned out to be effective for improving the reconstruction accuracy [1]. Shih et al. produced high-resolution depth maps of the scenes utilizing a multi-camera system, which consists of a wide-angle camera and an array of telescopic cameras [2]. However, dual-camera and multi-camera systems usually occupy a large spatial layout and have limitations in some scenarios. Different perspectives images can also be successively achieved by one moving camera. Then under the framework of structure from motion (SfM) [3] the spatial information of the scenes can be reconstructed from the images captured over time. For example, Gao et al. designed a set of multi-view 3D reconstruction technology based on SFM and obtained the external parameters of the camera and the sparse point-cloud model [4]. Wang et al. integrated the fringe light projection with SfM to realize a global dense point cloud reconstruction system [5]. These SfM-based methods can produce accurate 3D point clouds for large-scale scenes; however, capturing images one after another is time consuming and inconvenient for many applications. 

An alternative approach to 3D reconstruction is to introduce an optical light deflection device in front of the camera. For example, Zhang et al. proposed a system consisting of a single camera and a fixed mirror to reconstruct the 3D object [6]. The correspondence between the object and its counterpart in the mirror is auto-epipolar in one image taken by the camera; therefore, the object can be computed in a way similar to using a dual-camera system. Yu et al. proposed a camera–mirror-based stereo digital image correlation (DIC) system for 3D shape and deformation measurement. The four fixed mirrors in the DIC system divide the camera’s field of view (FOV) into two parts, which observe the object from two different angles and constitute a binocular 3D reconstruction system [7]. By utilizing fixed mirror(s), these methods effectively formed the geometrical constraints for 3D reconstruction. However, the image plane had to be divided into two or more parts in these methods to capture the stereo views simultaneously; thus, the effective FOV and the spatial resolution were much reduced. Recently, some adjustable light deflectors, such as the transparent rotating deflector (TRD) mounted on a stepping motor [8], the optical plate rotated with electrothermal MEMS actuator [9], and the rotatable wedge prism [10], have been utilized for changing the camera boresight. The optical model and the 3D reconstruction method based on these boresight-variable imaging systems have been extensively investigated. However, the reported 3D reconstruction accuracy results achieved with these boresight-variable imaging systems so far are much lower than that of the conventional stereopsis [9,10,11].

The galvanometer scanner is another type of light deflection device, which consists of two optical reflection mirrors fixed on their rotation shafts. It features high speed and high accuracy for light deflection [12,13] and has been widely applied in laser scanning for material processing [14,15], medical treatment [16,17], and laser indication [18,19]. The galvanometer scanner was also combined with the camera to enlarge the FOV of the vision system for surveillance [20], large-scale structure inspection [21], high-speed moving object tracking [22], etc. Owing to the ultrafast light defection capability, the galvanometer scanner was also applied in image capturing for motion blur compensation [23,24]. The above galvanometer–camera imaging systems (GCIS) mainly work for capturing clear detail images in a large FOV or close-up image tracking of fast-moving objects, not for 3D reconstruction. To enable the GCIS to be conveniently applied in 3D vision tasks as a conventional camera, we investigated the imaging model and the calibration method of the GCIS in an early work [25], in which a single hidden layer feedforward neural network was utilized to establish the imaging relation of the GCIS to incorporate the mirror-deflection-induced boresight change. Since the GCIS was regarded as a single imaging device with an enlarged FOV in Ref. [25], two or more GCISs are needed to reconstruct the 3D coordinate information of the spatial points. Some other galvanometer-based 3D reconstruction systems have also been proposed. Hegna et al. proposed a 3D reconstruction system combing a laser rangefinder and a galvanometer [26]. Shi et al. proposed a time of flight (ToF)-based laser–galvanometer 3D reconstruction system [27].

In this paper, we model the combination of one galvanometer and one camera as a virtual multi-ocular system (VMOS). The configuration and working mechanism with definite physical meaning are presented. A simple and efficient method for calibrating the intrinsic and extrinsic parameters of the virtual multi-ocular system is put forward. The applicability of the proposed system for 3D reconstruction and pose estimation is investigated, and the results demonstrate that the proposed method can achieve competitive accuracy with that of conventional multi-view stereopsis with a much more concise hardware configuration.

## 2. Methodology

### 2.1. Configuration and Construction of the VMOS

The configuration of the proposed galvanometer–camera combined VMOS is shown as Figure 1a. The VMOS consisted of a galvanometer scanner, a camera with an appropriate lens, and a set of control units. The galvanometer scanner was fixed in front of the camera, and the control unit was used to control the camera and the galvanometer scanner simultaneously to take pictures when the galvanometer deflected to a specified position. The lights of the scene were deflected twice by the two mirrors in the galvanometer scanner and then captured by the camera sensor through the lens. By changing the turning angles of the two mirrors, the camera boresight and FOV could be adjusted, as shown in Figure 1b.

According to the principle of mirror transformation, changing the camera’s field of view through the mirror deflections is equivalent to changing the camera’s pose (including the position and direction), as shown in Figure 2.

In Figure 2, O−XYZ is the camera coordinate system, which represents the pose of the real camera. The rotation angles of *Mirror*-1 and *Mirror*-2 are denoted as α and β, respectively, which are uniquely determined by a pair of control values D(a,b). Suppose *Mirror*-1 and *Mirror*-2 are at the initial turning angles, then $O_{1}^{\prime }$ is the virtual camera position, which is specularly transformed from O with *Mirror*-1, and $O_{1}^{\prime \prime }$ is the virtual camera position, which is specularly transformed from $O_{1}^{\prime }$ with *Mirror*-2 in the initial status. The boresight of the real camera is identically transformed and marked as the blue dotted lines. When the turning angles α and β are changed to an arbitrary status, the corresponding virtual camera positions induced by the two mirror transformations are denoted as $O_{2}^{\prime }$ and $O_{2}^{\prime \prime }$, respectively, and the virtual camera boresight in this status is marked as the red dotted lines. 

To sum up, the virtual camera pose was related to the deflection angles α and β, the distance between the rotation axes of the two mirrors, and the relative installation pose between the real camera and the galvanometer scanner. However, it is not trivial to directly calculate the pose matrices of the virtual cameras in practice for the following reasons: (1) The turning angles α and β are determined by a pair of control parameters a and b, respectively. The non-linear mapping between (α,β) and the control parameter D(a,b) needs to be carefully calibrated, and the calibration errors of D(a,b)→(α,β) may reduce the accuracy of the calculated virtual camera poses. (2) The distance between the rotation axes of two mirrors is determined by the manufacturing process of the galvanometer scanner and is difficult to accurately measure in practice. (3) The relative installation pose between the camera and the galvanometer scanner is hard to know.

Instead of trying to calculate the virtual camera poses through specular reflection transformation, we enabled the galvanometer–camera to work as a virtual multi-ocular system, which needed to know neither the nonlinear relation D(a,b)→(α,β), the rotation axes distance of the two mirrors, nor the installation pose of the camera. This scheme took advantage of the high repeatability of the galvanometer scanner. Specifically, the high repeatability of the scanner meant that whenever a specific control parameter D(a,b) was transmitted to the scanner, the corresponding deflection angles α and β almost remained unchanged every time, and hence, the imaging area of the system was all the same. In other words, given control parameter D(a,b), the pose of the virtual camera V was definitely determined. Therefore, we sampled the 2D control parameter domain in advance and endeavored to calibrate the corresponding virtual poses that corresponded to the sampled parameters. A one-to-one mapping D→V from the sampled control parameters D to the corresponding virtual camera V was established. All the virtual cameras constituted the virtual multi-ocular system. 

In order to perform the camera imaging within the deflection range of the galvanometer scanner, the camera and the galvanometer scanner should be properly configured to guarantee that the view pyramid of any virtual camera resulting from the deflection of *Mirror* -1 should intersect with *Mirror*-2, as shown in Figure 3.

More specifically, the parameters of the galvanometer–camera combination should meet the following condition:(1)$$\cot(2\alpha -\frac{\theta }{2})(\sin2\alpha *OR_{1}+R_{1}R_{2})-\cos2\alpha *OR_{1}\leq \frac{W}{2},$$
where α is the turning angle of *Mirror*-1, θ is the FOV angle of the camera, *W* is the width of *Mirror*-2, O is the optical center point of camera, $O^{\prime }$ is the optical center point of virtual camera formed by *Mirror*-1, $R_{1}$ is the center point of *Mirror*-1, and $R_{2}$ is the center point of *Mirror*-2.

To guarantee that each virtual camera in the VMOS shared common FOVs with some of the others, the sampling numbers of control parameter D(a,b) should satisfy
(2)$$\left\{\begin{array}{l} N_{a}\geq \frac{2\alpha_{\max}}{\theta_{y}/2}\\ N_{b}\geq \frac{2\beta_{\max}}{\theta_{x}/2} \end{array}\right.,$$
where $N_{a}$ and $N_{b}$ are the least sampling numbers of the control parameters a and b, respectively; $\alpha_{\max}$ and $\beta_{\max}$ are the maximum turning angles of *Mirror*-1 and *Mirror*-2, respectively; and $\theta_{x}$ and $\theta_{y}$ are the camera FOV angle in the horizontal and vertical directions, respectively. Having determined $N_{a}$ and $N_{b}$, the 2D control parameter domain is evenly sampled. Then we have a number of $S=N_{a}\times N_{b}$ virtual cameras corresponding to the sampled control parameters $D{\left(a,b\right)}_{s}(s=1,2,\ldots ,S)$. The virtual cameras are denoted as $V_{s}(s=1,2,\ldots ,S)$.

The above control parameter sampling rule can ensure that the adjacent virtual cameras share common FOVs. Most viewable regions of the VMOS largely have fourfold overlap, as shown in Figure 4. The bigger the sampling numbers $N_{a}$ and $N_{b}$ are, the more folds the viewing regions overlap, and the more constraints can be supplied for 3D reconstruction. 

### 2.2. Calibration Method of the VMOS

According to Section 2.1, the VMOS was composed of a number of $S=N_{a}\times N_{b}$ virtual cameras corresponding to the sampled control parameters $D{\left(a,b\right)}_{s}(s=1,2,\ldots ,S)$. Since all the virtual cameras were induced from the same real camera, the intrinsic parameters, including the pinhole imaging matrix and the distortion parameters, were the same for each virtual camera, while the poses of all the virtual cameras $V_{s}(s=1,2,\ldots ,S)$ needed to be calculated.

Due to the large FOV of the VMOS, the calibration was difficult to realize by a calibration target at once. We proposed a global optimization method for zonal calibration, combining Zhang’s camera calibration method [28], the PnP (perspective-n-point) method [29], and the bundle adjustment (BA) method [30]. The main steps are summarized in Figure 5.

To realize the calibration method, we built a planar calibration target, which was evenly distributed with coded points. The identifications of each coded points in the images could be easily recognized by decoding. Denote the calibration target coordinate system as *C-CS*, and the coordinates of the coded points in *C-CS* are denoted as $X^{id}$, where the superscript id represents the identification of a coded point. The specific steps of proposed calibration method are as follows:

For image collection and calibration data preparation, put the calibration target at position $T_{p}(p=1,2,\ldots ,P)$ in the working volume of the VMOS. Capture images $I_{sp}(s=1,2,\ldots ,S,p=1,2,\ldots ,P)$ of the target in position $T_{p}$ with the virtual camera $V_{s}$. Then extract the image coordinates $x_{sp}^{id}$ of the coded points in image $I_{sp}$. The 3D coordinates under a global coordinate system (*G-CS*) of the coded points $X_{p}^{id}$ on the calibration target in position $T_{p}$ are measured utilizing a photogrammetric device.

For the calibration of the camera intrinsic parameters, among images $I_{sp}$ with different index s and fixed index p, match the 3D points $X_{p}^{id}$ with the image points $x_{sp}^{id}$. Take the matched pairs $X_{p}^{id}↔x_{sp}^{id}$ into Zhang’s monocular camera calibration process [28,31] for calibrating the intrinsic matrix K in the pinhole camera model as shown in Equation (3) and the distortion parameters $\left(k_{1},k_{2},k_{3},k_{4},k_{5}\right)$ expressed in Equation (4).
(3)$$\lambda \left[\begin{array}{c} u\\ v\\ 1 \end{array}\right]=K[R|t]\left[\begin{array}{c} x\\ y\\ z\\ 1 \end{array}\right]$$
where ${\left[x,y,z,1\right]}^{T}$ is the homogeneous coordinates of the spatial point, ${\left[u,v,1\right]}^{T}$ is the ideal homogeneous pixel coordinates of the corresponding point, [R|t] is the pose parameters of the camera, and λ is the depth coefficient.
(4)$$\left\{\begin{array}{l} u_{d}=u+u(k_{1}r^{2}+k_{2}r^{4}+k_{3}r^{6})+[2k_{4}uv+k_{5}(r^{2}+2u^{2})]\\ v_{d}=v+v(k_{1}r^{2}+k_{2}r^{4}+k_{3}r^{6})+[k_{4}(r^{2}+2v^{2})+2k_{5}uv] \end{array}\right.,$$
where $u_{d}$ and $v_{d}$ are the observed pixel coordinates with distortion corresponding to the ideal coordinates u and v, respectively; r is the distance between the pixel point (u,v) and the principle point of the pixel plane; $k_{1}$, $k_{2}$, and $k_{3}$ are the radial distortion parameters; and $k_{4}$ and $k_{5}$ are the tangential distortion parameters.For the calibration of the virtual camera poses, to calculate the *s*th virtual camera pose, gather the coded points $x_{sp}^{id}$ in the images $I_{sp}$ as a group $G_{s}$ with the same index s(s=1,2,…,S) and different index p. Match the image points $x_{sp}^{id}$ in each $G_{s}$ with $X_{p}^{id}(p=1,2,\ldots ,P)$ according to index p and id. Utilizing the matched pairs $X_{p}^{id}↔x_{sp}^{id}$ in the specific group $G_{s}$, the pose of the virtual camera $V_{s}$(s=1,2,…,S), i.e., the transformation matrix $\left[R_{s}|t_{s}\right]$ from *G-CS* to the virtual camera coordinate system $V_{s}$*-CS*, is calculated through the PnP method [29]. For global optimization, to improve the calibration accuracy, the BA method [32] is applied to optimize the intrinsic parameters and all the virtual camera poses. In consideration of the lens distortion, we add radial distortion and tangential distortion to the BA model. The objective function of the nonlinear optimization is
(5)$$\delta (K,[R_{s}|t_{s}],k_{1},k_{2},k_{3},k_{4},k_{5})=\sum_{p=1}^{P}\sum_{s=1}^{S}\sum_{}^{id}{\left\|reprojection_{sp}^{id}-x_{sp}^{id}\right\|}_{2},$$
where $reprojection_{sp}^{id}$ is the reprojection pixel coordinates of spatial point $X_{p}^{id}$ in virtual camera $V_{s}$ calculated through Equations (3) and (4).

Figure 6 shows the schematic diagram of the entire calibration process. Finally, the intrinsic matrix K, the distortion parameters $\left(k_{1},k_{2},k_{3},k_{4},k_{5}\right)$, and the extrinsic matrices $\left[R_{s}|t_{s}](s=1,2,\ldots ,S\right)$ of the virtual cameras were determined.

### 2.3. The 3D Reconstruction Method with the VMOS

Having completed the VMOS calibration, the intrinsic matrix K, the distortion parameters $\left(k_{1},k_{2},k_{3},k_{4},k_{5}\right)$, and the extrinsic pose matrices $\left[R_{s}|t_{s}\right]$ of all the virtual cameras $V_{s}(s=1,2,\ldots ,S)$ were obtained. The control parameter sampling rule described in Section 2.1 guarantees that the scene in the working volume of the VMOS can be observed by largely four or more virtual cameras. According to the triangulation method, the region observed by multiple virtual cameras can be 3D reconstructed, shown as Figure 7.

In Figure 7, the image point $m_{1}$ corresponding to the spatial point *M* can be expressed as
(6)$$\lambda_{1}\left[\begin{array}{c} u_{1}\\ v_{1}\\ 1 \end{array}\right]=K[R_{1}|t_{1}]\left[\begin{array}{c} X\\ Y\\ Z\\ 1 \end{array}\right],$$
where $\left[R_{1}|t_{1}\right]$ is the 3×4 extrinsic pose matrix of $V_{1}$; ${\left[X,Y,Z,1\right]}^{T}$ is the homogeneous coordinates of *M* in the world coordinate system; ${\left[u_{1},v_{1},1\right]}^{T}$ is the undistorted pixel coordinates of *M* in pixel coordinate system of virtual camera $V_{1}$, which can be calculated from the observed image coordinates with Equation (4); and $\lambda_{1}$ is the depth coefficient of point *M* in the coordinate system of virtual camera $V_{1}$. By eliminating $\lambda_{1}$, Equation (6) can be reorganized as
(7)$$\left[\begin{array}{l} u_{1}{K[R_{1}|t_{1}]}^{\left(3\right)}-{K[R_{1}|t_{1}]}^{\left(1\right)}\\ v_{1}{K[R_{1}|t_{1}]}^{\left(3\right)}-{K[R_{1}|t_{1}]}^{\left(2\right)} \end{array}\right]\left[\begin{array}{c} X\\ Y\\ Z\\ 1 \end{array}\right]=0,$$
where ${\left[R_{1}|t_{1}\right]}^{\left(i\right)}(i=1,2,3)$ represents the *i*th row of matrix $\left[R_{1}|t_{1}\right]$.

According to Equation (7), one camera can provide a 2 × 4 coefficient matrix. When there are n(n≥2) cameras having observed the target point, an overdetermined linear system shown in Equation (8) can be obtained:
(8)$$\left[\begin{array}{c} u_{1}{K[R_{1}|t_{1}]}^{\left(3\right)}-{K[R_{1}|t_{1}]}^{\left(1\right)}\\ v_{1}{K[R_{1}|t_{1}]}^{\left(3\right)}-{K[R_{1}|t_{1}]}^{\left(2\right)}\\ u_{2}{K[R_{2}|t_{2}]}^{\left(3\right)}-{K[R_{2}|t_{2}]}^{\left(1\right)}\\ v_{2}{K[R_{2}|t_{2}]}^{\left(3\right)}-{K[R_{2}|t_{2}]}^{\left(2\right)}\\ \vdots \\ u_{n}{K[R_{n}|t_{n}]}^{\left(3\right)}-{K[R_{n}|t_{n}]}^{\left(1\right)}\\ v_{n}{K[R_{n}|t_{n}]}^{\left(3\right)}-{K[R_{n}|t_{n}]}^{\left(2\right)} \end{array}\right]\left[\begin{array}{c} X\\ Y\\ Z\\ 1 \end{array}\right]=0.$$


Perform singular value decomposition (SVD) [30] on the coefficient matrix. The 3D coordinates (X,Y,Z) can be obtained from the singular vector corresponding to the minimum singular value. 

### 2.4. Pose Estimation Method Using the VMOS

Object pose estimation is one of the most common applications of machine vision. The PnP algorithm is the most common means of monocular pose estimation [33,34]. Through the 2D image coordinates observed by one camera and the corresponding known 3D coordinates of the target, the transformation between the object coordinate system and the camera coordinate system can be calculated. Then the six degree of freedom (DOF) pose parameters of the object with respect to the camera coordinate system can be obtained.

However, due to the limited field of view of each virtual camera, it may be impossible to obtain enough points for the PnP calculation in a single perspective. In addition, the pose calculated by PnP from a single view is in the current virtual camera coordinate system, which needs to be converted to the VMOS coordinate system using the pose parameters of each virtual camera, which is cumbersome and inconvenient. Fortunately, the proposed VMOS could observe the same object point by different virtual cameras, which had potential to provide more constraints for determining the object pose compared with ordinary cameras. Taking advantage of the large FOV of the VMOS, we proposed a global pose estimation algorithm to directly obtain the object pose in the VMOS coordinate system by utilizing the images from multiple virtual cameras.

In our pose estimation scheme with the VMOS, not all the virtual cameras but only those having observed the feature points for the pose estimation participated in the calculation. As shown in Figure 8, suppose the calibrated virtual camera $V_{i}$ observes a point $P_{j}$ in the object coordinate system (*O-CS*) concerning the pose estimation, and the corresponding undistorted pixel coordinates $\left(u_{ij},v_{ij}\right)$ are obtained. Then, $\left(u_{ij},v_{ij}\right)$ can be transformed to the normalization plane in $V_{i}-CS$ according to (9).
(9)$$\left[\begin{array}{c} u_{ij}\\ v_{ij}\\ 1 \end{array}\right]=K\left[\begin{array}{c} x_{ij}\\ y_{ij}\\ 1 \end{array}\right],$$
where ${\left[x_{ij},y_{ij},1\right]}^{T}$ is the coordinates of point $p_{ij}$, which is on the normalization plane in $V_{i}-CS$ corresponding to $P_{j}$.

The correspondence between spatial point $P_{j}$ and the line $O_{i}p_{ij}$ was established. Utilizing the extrinsic matrix $\left[R_{i}|t_{i}\right]$, the line $O_{i}p_{ij}$ was transformed from $V_{i}-CS$ to the VMOS coordinate system (i.e., G−CS), as shown in Equations (10) and (11).
(10)$${\vec{l}}_{ij}^{G}=R_{i}^{-1}{\vec{l}}_{ij}$$
(11)$$O_{i}^{G}=-R_{i}^{-1}t_{i}$$
where ${\vec{l}}_{ij}^{G}$ is the normalized orientation vector of line $O_{i}p_{ij}$ in *G-CS*, $O_{i}^{G}$ is a passing point of line $O_{i}p_{ij}$ in *G-CS*, and ${\vec{l}}_{ij}$ is the normalized orientation vector of $O_{i}p_{ij}$ in $V_{i}-CS$,
(12)$${\vec{l}}_{ij}=\frac{\left(x_{ij},y_{ij},1\right)}{\left\|\left(x_{ij},y_{ij},1\right)\right\|}.$$

Given *N* pairs of 3D point–line correspondence which formed by virtual cameras $V_{i}(i=1,2,\ldots ,I)$ observing points $P_{j}(j=1,2,\ldots ,J)$ on an object, the pose estimation can be modeled as the non-perspective PnP (NPnP) [35] problem depicted as
(13)ζijl→ijG+OiG=ROGPj+tOG for i=1,2,…,I; j=1,2,…,J
where $\zeta_{ij}$ is the parameter of line $O_{i}p_{ij}$ and [OGRtOG] is the transformation matrix from *O-CS* to *G-CS.*

We utilized the procrustean solution provided in [35] to estimate the transformation matrix [OGRtOG] in Equation (13). After obtaining the result, a BA optimization was performed to improve the accuracy. Taking [OGRtOG] as the initial value, we minimized the reprojection errors expressed in Equation (14) to finally obtain the object pose parameters.
(14)δ(OGRtOG)=∑j∑i(u˜ij,v˜ij)−(uij,vij)2
where $\left({\tilde{u}}_{ij},{\tilde{v}}_{ij}\right)$ is the reprojection pixel coordinates of the spatial point $P_{j}$ in virtual camera $V_{i}$ and can be expressed in Equation (15).
(15)u˜ijv˜ij1=KRi|tiOGRtOGXjYjZj1.

The Gauss–Newton iteration method was used to minimize the objective function in Equation (14).

## 3. Experiments

### 3.1. Hardware Setup

The VMOS used in our experiments is shown in Figure 1a. Specifically, the hardware contained a CCD camera with a lens and a galvanometer with two reflection mirrors. The camera used was an GT-2450 from AVT in Germany, whose resolution is 2448 × 2050 pixels, and the size of each pixel is 3.45 µm × 3.45 µm. The lens installed on the camera was a LM50-JC1MS from Kowa in Japan with a 50 mm fixed focal length. The nominal repeatability of the used galvanometer was <8 µrad. The planar calibration target used in the experiments was a piece of glass, 2.4 m × 1.2 m in size and with a set of coded markers [36] pasted on, as shown in Figure 9. Each coded marker had eight white circular dots distributed in a specific pattern on a black background. From the unique distribution pattern of each marker, the marker identity and in turn the identity of each white dot could be recognized. This helped to establish the correspondences between the points on the board and the points in the images. The program for completing the whole experimental process was written in C++ and installed in a personal computer with an Core i5-9400F @3.7 GHz CPU from Intel in America, 16 G RAM, and a Windows 10 Enterprise edition operating system from Microsoft in America.

### 3.2. Calibration Experiment

#### 3.2.1. Galvanometer Repeatability Verification

Since the proposed VMOS calibration method was based on the high repeatability characteristic of the galvanometer scanner, we first conducted an experiment to verify the repeatability of the galvanometer.

We placed the VMOS about 2.5 m in front of a wall pasted with nine coded markers, as shown in Figure 10. The galvanomirrors in the VMOS were deflected so that the camera could observe each of the coded markers, and the corresponding control parameters of the galvanometer were recorded as $D{\left(a,b\right)}_{s}(s=1,2,\ldots ,9)$. We then repeatedly input the recorded $D{\left(a,b\right)}_{s}(s=1,2,\ldots ,9)$ to the VMOS five times and captured five images for each coded marker. A total of 45 images were captured.

The pixel coordinates of the coded point centers in the images were extracted. The deviations among the five pixel coordinates of the same coded point center corresponding to the same $D{\left(a,b\right)}_{s}$ were calculated. The mean absolute (MA) and root mean square (RMS) deviations in the pixel are shown in Table 1.

The repeatability of galvanometer could be calculated by Equation (16).
(16)$$\delta =\arctan(\frac{\varepsilon \cdot p}{f}),$$
where *δ* is the repeatability of the galvanometer, *ε* is the coded point center deviation in pixel, *p* is the pixel size of the camera sensor, and *f* is the focal length of the camera lens.

According to the results in Table 1, the repeatability of the galvanometer calculated by Equation (16) was 4.62 µrad, which conformed with the nominal value.

#### 3.2.2. VMOS Calibration

After verifying the galvanometer repeatability, a calibration experiment was conducted. The camera’s horizontal and vertical viewing angles $\theta_{x}$ and $\theta_{y}$ were 9.6° and 8.0°, respectively. The maximum turning angles of the galvanomirrors were both about 20°. Referring to Equation (2), the sampling numbers $N_{a}$ and $N_{b}$ of the control parameter domain D(a,b) should be larger than 10 and 9, respectively, to satisfy the fourfold FOV overlap constraint. We set $N_{a}=N_{b}=21$ in the experiments. Therefore, 21 × 21 = 441 control parameter samples were obtained. The angle between adjacent virtual camera boresights was about 2°.

The measurement device used for determining the 3D coordinates of the dot centers on the calibration board was an industrial photogrammetry system TriTop^®^ from GOM Co. in Brunswick, Germany. It needed only a handheld camera with a fixed focal length to take a set of overlapped images of the calibration board from various perspectives to reconstruct the 3D coordinates of the white dot centers. Some coded points placed in the calibration scene were used to establish the global coordinate system *G-CS*.

The calibration operation processes in the experiment were as follows.

Put the planar calibration target in six positions, i.e., $T_{p}(p=1,2,\ldots ,6)$. Start the VMOS and take images of the target in each position with the 441 virtual cameras $V_{s}(s=1,2,\ldots ,441)$. Measure the coordinates $X_{p}^{id}$ of the coded points on the calibration target with TriTop^®^. The measured coded points on the calibration board in the six positions in *G-CS* are shown in Figure 11.

We chose the calibration target position $T_{1}$ to calibrate the intrinsic matrix K and the distortion parameters $\left(k_{1},k_{2},k_{3},k_{4},k_{5}\right)$ following the method described in step 1 in Section 2.2. Then, following step 2 and step 3 in Section 2.2, the transformation matrices $\left[R_{s}|t_{s}\right]$ of the virtual camera $V_{s}$(s=1,2,…,441) were computed and optimized, and the mapping $D{\left(a,b\right)}_{s}\to \left[R_{s}|t_{s}\right]$(s=1,2,…,441) was established. Some of the calibrated virtual cameras are shown in Figure 12. The arrow lines represent the boresights of the virtual camera, and the starting points of the lines are the optical centers of the virtual cameras.

We used the reprojection errors of all the spatial coded points on all the virtual cameras to evaluate the calibration accuracy. The MA and RMS of the reprojection errors are listed in Table 2, where the “direction” column includes the error components in the horizontal and vertical directions, as well as the full reprojection error.

According to the calibration results shown in Table 1, the optimized parameters K, $\left(k_{1},k_{2},k_{3},k_{4},k_{5}\right)$, and $\left[R_{s}|t_{s}\right]$(s=1,2,…,441) resulted in about a 5% reduction in the MA reprojection error and about 10% in the RMSE compared with the initial values. The comparison revealed that the optimization process played an important role in improving the calibration accuracy. In this experiment, the size of the calibration target covered by each image was about 500 mm × 400 mm. Owing to the relatively high image resolution (2448 × 2050), the absolute length represented by 1 pixel was about 0.2 mm in the calibration target area. After the optimization, the mean reprojection error was 0.282 pixel, which largely stood for an accuracy of about 0.06 mm in space.

### 3.3. Experiments on 3D Coordinate Reconstruction

We performed two 3D reconstruction experiments using the calibrated VMOS to verify its applicability in 3D vision tasks.

#### 3.3.1. Reconstruction of a Visual Scale Bar

The visual scale bar is commonly used in photogrammetry to serve as a scale reference. It has visual markers with precise known distances between each other. In this experiment, we reconstructed the 3D coordinates of the 24 white dot centers belonging to three coded markers on a scalar bar to verify the 3D reconstruction capability of the VMOS, as shown in Figure 13. The distance between the VMOS and the scale bar was about 2.5 m.

Following the method in Section 2.3, the coordinates of the 24 points on the three coded markers on the scale bar were reconstructed with the calibrated VMOS, as shown in Figure 14. To evaluate the reconstruction accuracy, the coordinates of the three sets of coded points were measured using TriTop^®^ from GOM CO. in Brunswick, Germany as well and regarded as the ground truth. Then, the distances between each dot of the No. 0 coded marker and each dot of the No. 1 and No. 2 coded markers were calculated. There were 128 distances calculated in total, which were compared with the ground truth values to evaluate the 3D reconstruction. The absolute mean (AM) and root mean square (RMS) distance errors are shown in Table 3.

#### 3.3.2. Reconstruction of Marker Points on 3D Structure

In this experiment, we utilized the VMOS to reconstruct a 3D structure object. We placed the object in the working space of the VMOS, shown in Figure 15.

Referring to the method in Section 2.3, the coordinates of 19 × 8 coded points on the 3D object surface were reconstructed, as shown in Figure 16. In order to verify the reconstruction accuracy, the reconstructed 3D coordinates of the coded points were compared with the coordinates measured by TriTop^®^. The MA and RMS reconstruction errors of the 19 × 8 coded points scattered on the 3D structure are listed in Table 4, including the error components in the x, y, and z directions and the full Euclidean distance errors.

The results in Table 4 show that the errors in the x and y directions were much smaller than that in z direction (i.e., the depth direction). The reason was that the baseline between each virtual camera was much smaller than the target depth, resulting in the uncertainty increase in the depth direction. The related works on 3D reconstruction using a variable boresight imaging system [10,37] also reported relatively lower accuracy in the depth direction. In our experiment, we quantitatively evaluated the MA and RMS reconstruction errors of the coded points, which could be accurately measured to serve as the true values.

#### 3.3.3. Repeatability of Reconstruction Verification

In order to test the repeatability of the VMOS reconstruction, we conducted a repeatability verification experiment. We placed the 3D structure object as shown in Figure 15 at a location in the FOV of the VMOS and reconstructed the coded points on the object by using the VMOS. Then we changed the position of the object and reconstructed the object coded points again. The processes were repeated, and five sets of reconstruction results were obtained, as shown in Figure 17.

To evaluate the repeat accuracy of the 3D reconstruction, we aligned the reconstructed point sets under each position to the other sets, and 10 pairs of aligned point sets were obtained. The distance errors of the corresponding points in each pair of point sets after the alignment were calculated, and the errors are shown in Table 5. The experiment results demonstrated a good repeatability of the VMOS.

### 3.4. Experiment on Pose Estimation 

In this experiment, we demonstrated the applicability of the calibrated VMOS for two objects’ relative pose estimations, which is a frequently faced 3D vision task in many scenarios. Two 3D objects, denoted as *Obj A* and *Obj B*, were placed in the FOV of the calibrated VMOS, as shown in Figure 18.

We denoted the coordinate systems of the objects as $O_{a}-CS$ and $O_{b}-CS$. The relative pose between $O_{a}-CS$ and $O_{b}-CS$ was unknown and needed to be determined. There were some coded markers on the surface of each object. In order to obtain a ground truth of the relative pose to evaluate the estimate result, we again used the TriTop^®^ to measure the coded points on *Obj A* and *Obj B* and then calculated the relative pose transformation matrix $T_{AB}$ as the ground truth.

There were 174 out of the total 441 virtual cameras participating in the pose estimation in this experiment, as shown in Figure 19a, where dark gray cells indicate the corresponding sampled control parameters. By using the global pose estimation method in Section 2.4, the transformation matrix $T_{AG}$ from $O_{a}-CS$ to G−CS and transformation matrix $T_{BG}$ from $O_{b}-CS$ to G−CS were calculated. As shown in Figure 19b, the blue and orange circles represent the points on *Obj A* and *Obj B*, respectively. The blue and orange lines represent the calculated spatial lines in *G-CS* corresponding to the image points on *Obj A* and *Obj B* captured by the virtual cameras. The transformation matrix from *Obj A* to *Obj B* obtained using the VMOS was denoted as $T_{AB}^{G}$, which was calculated by Equation (17).
(17)$$T_{AB}^{G}=T_{BG}{}^{-1}T_{AG}$$

For comparison, we also estimated the relative pose of *Obj A* to *Obj B* using an ordinary camera, as shown in Figure 20. The model of the ordinary camera was GC-2450 from AVT in Germany, which had the same performance parameters as the camera used in the VMOS. The lens installed on the camera was a LM12-JC1MS from Kowa in Japan with a 12 mm focal length, which was the same series with the lens installed in the VMOS. This lens enabled the ordinary camera to obtain a similar FOV to that of the VMOS. 

In the ordinary camera pose estimation experiment, we took a PnP solver (SQPnP) [38], which was based on the non-linear sequential quadratic programming method, to calculate the transformation matrix $T_{AC}$ and $T_{BC}$ from $O_{a}-CS$ and $O_{b}-CS$ to the camera coordinate system, respectively. Similarly, the transformation matrix from *Obj A* to *Obj B* obtained using the ordinary camera is denoted as $T_{AB}^{C}$, which was calculated by Equation (18).
(18)$$T_{AB}^{C}=T_{BC}{}^{-1}T_{AC}$$

According to the transformation matrices $T_{AB}^{}$, $T_{AB}^{G}$, and $T_{AB}^{C}$, the 6D pose parameters (rotations around the x, y, and z axes and translations along the x, y, and z directions) were calculated. The errors in the parameters obtained by the VMOS and those obtained by the ordinary camera were compared with the ground truth provided by TriTop^®^, as shown in Table 6.

The results in Table 6 illustrate that the errors in the z-direction (depth direction), as well as the overall errors, of the VMOS were much lower than those of the ordinary camera. The VMOS outperformed the ordinary camera in the relative pose estimation. One reason might have been that the multiple virtual cameras of the VMOS provided more constraints, which was conducive to eliminating the uncertainty error in the pose estimation compared with the single ordinary camera. In addition, the VMOS had a much higher resolution than the ordinary camera for the same FOV and was helpful in finding the exact image location of the coded points.

## 4. Conclusions

A novel 3D reconstruction system, which only consisted of a galvanometer scanner and a single camera with lens, is introduced. The galvanomirrors deflect the incoming rays and induce multiple virtual cameras in different poses. A set of methods for calibrating the intrinsic and extrinsic parameters of the virtual multi-ocular system is presented. The calibrated VMOS is applicable for 3D reconstruction and 6D pose estimation. The experimental results quantitatively demonstrate that the proposed VMOS can achieve 3D reconstruction with the accuracy of about 1 mm. Moreover, the VMOS shows better performance than the ordinary camera in pose estimation. Compared with the traditional multi-ocular system, the VMOS avoids the bulky hardware and complex layout, while achieving a comparable 3D reconstruction performance. Compared with the SfM-based methods, the ultrafast scanning capability enables the VMOS to swiftly capture the target images, which is crucial in many applications.

One focus of future work may be reconfiguring the hardware setup to enlarge the baselines between the virtual cameras in the VMOS, which is promising for further improvement in the reconstruction accuracy. It can be realized by increasing the distance between the camera and the galvanometer scanner under the constraints of Equation (1). In addition, the VMOS is developed for industrial inspection applications at present, which usually concern indoor settings and cooperative visual targets. However, it has potential to be used in non-laboratory environments. Application scenarios in more challenging surroundings can be explored in the future.

## Figures and Tables

**Figure 1 sensors-23-03499-f001:**
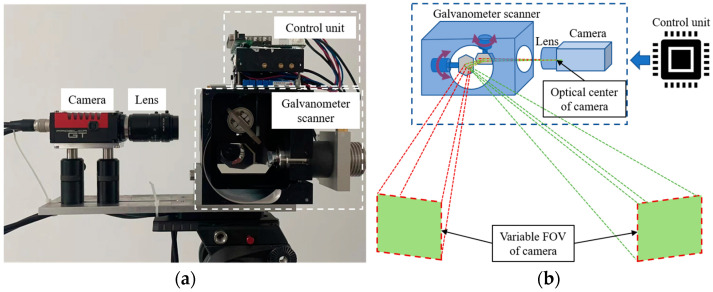
Diagram of the VMOS. (**a**) Configuration of the VMOS. (**b**) The schematic of the VMOS adjusting the boresight and FOV of the camera.

**Figure 2 sensors-23-03499-f002:**
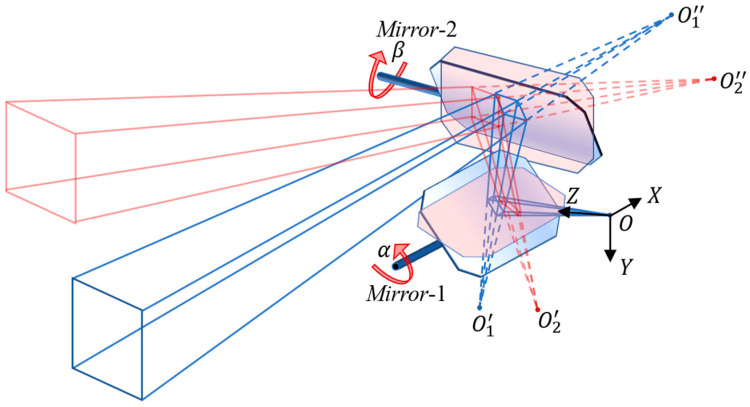
The virtual camera pose transformed by specular reflection transformation.

**Figure 3 sensors-23-03499-f003:**
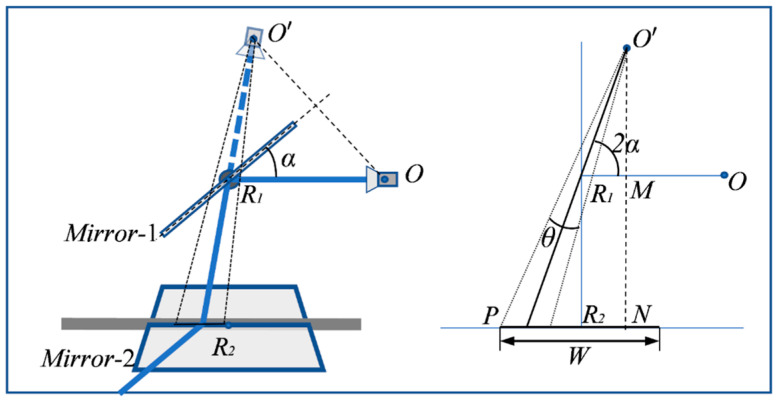
Camera imaging through galvanometer scanner.

**Figure 4 sensors-23-03499-f004:**
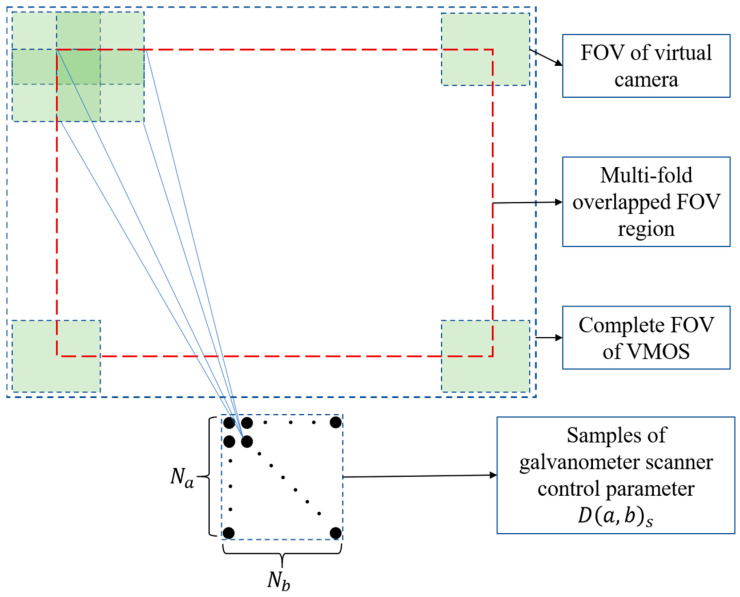
The control parameter samples and the virtual cameras’ overlapped FOVs.

**Figure 5 sensors-23-03499-f005:**
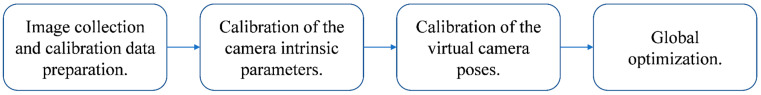
The main steps of the VMOS calibration.

**Figure 6 sensors-23-03499-f006:**
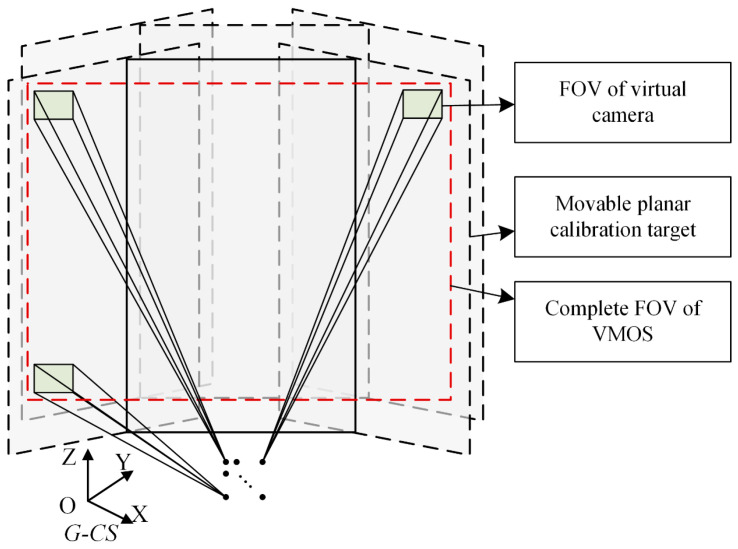
Schematic diagram of the calibration process.

**Figure 7 sensors-23-03499-f007:**
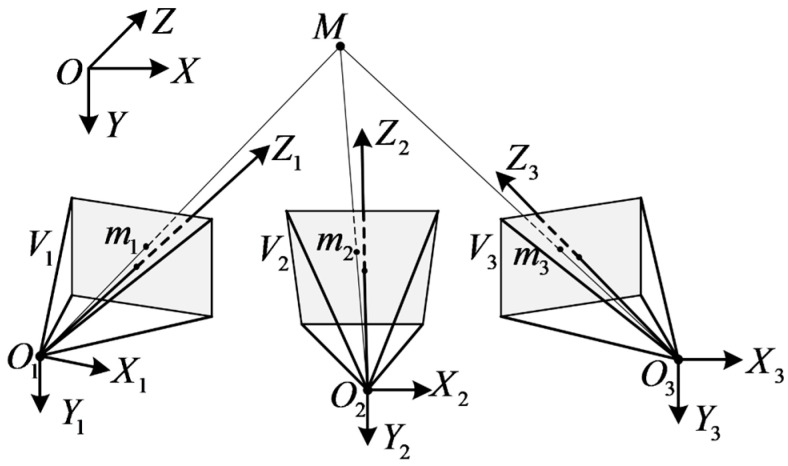
Principle of multi-ocular reconstruction.

**Figure 8 sensors-23-03499-f008:**
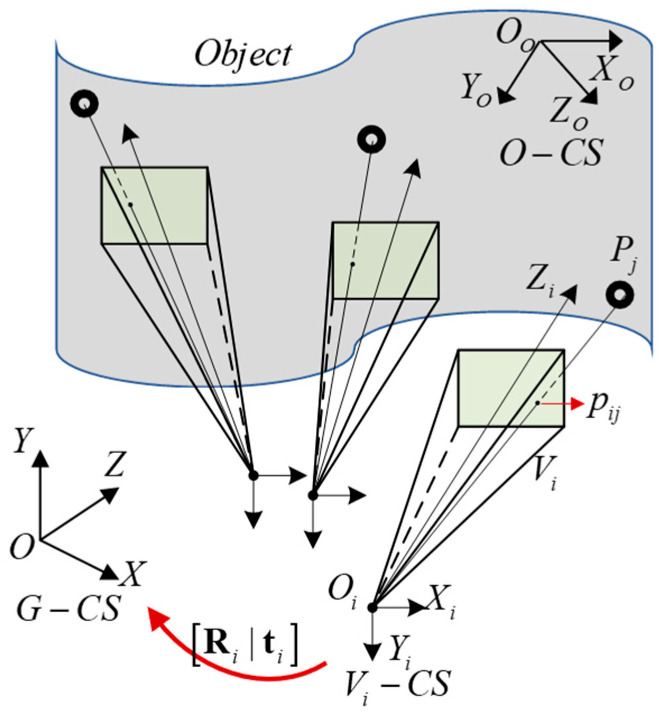
Correspondence of spatial point and line in the VMOS pose estimation.

**Figure 9 sensors-23-03499-f009:**
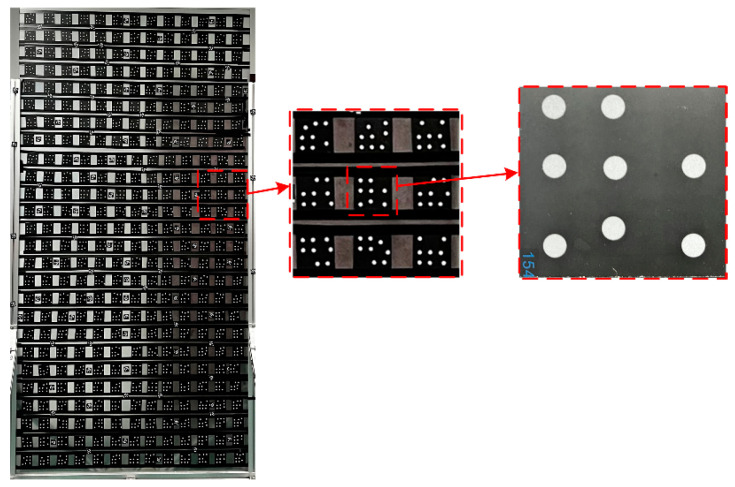
The planar calibration target used in the experiments.

**Figure 10 sensors-23-03499-f010:**
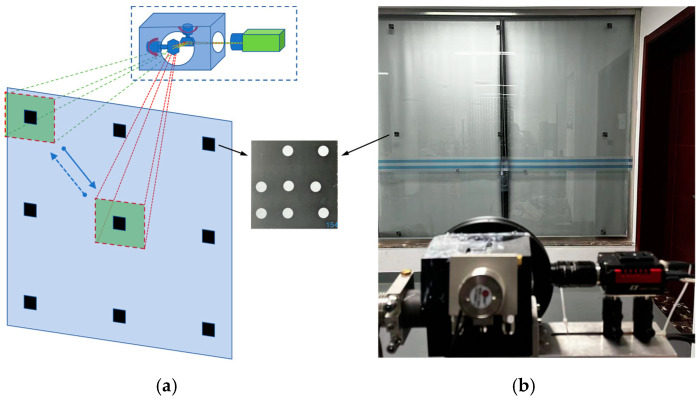
The experiment of galvanometer scanner repeatability. (**a**) The schematic diagram of the galvanometer repeatability experiment. (**b**) The scene of the repeatability experiment.

**Figure 11 sensors-23-03499-f011:**
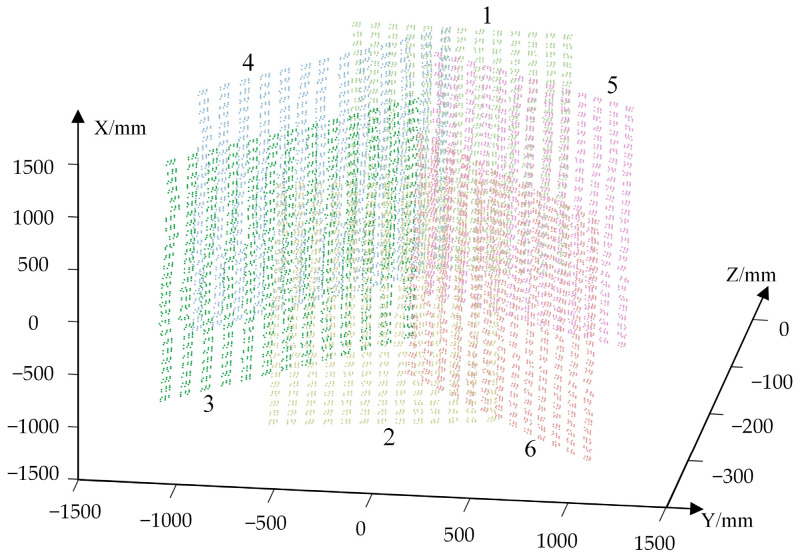
The coded points on the calibration target in six positions in *G-CS*.

**Figure 12 sensors-23-03499-f012:**
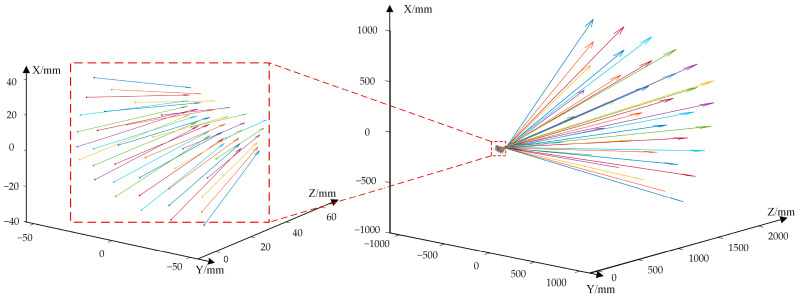
The distribution of partial virtual camera boresights.

**Figure 13 sensors-23-03499-f013:**
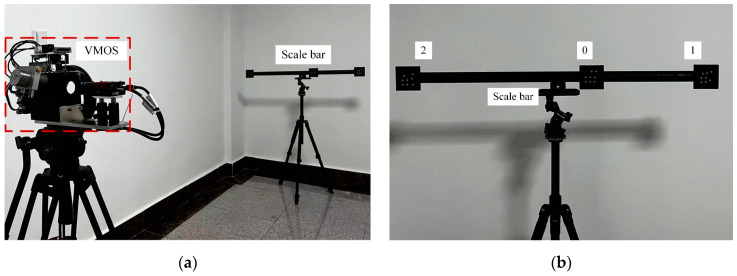
Scene of the scale bar reconstruction experiment. (**a**) The experiment scene. (**b**) Close-up of the scale bar.

**Figure 14 sensors-23-03499-f014:**
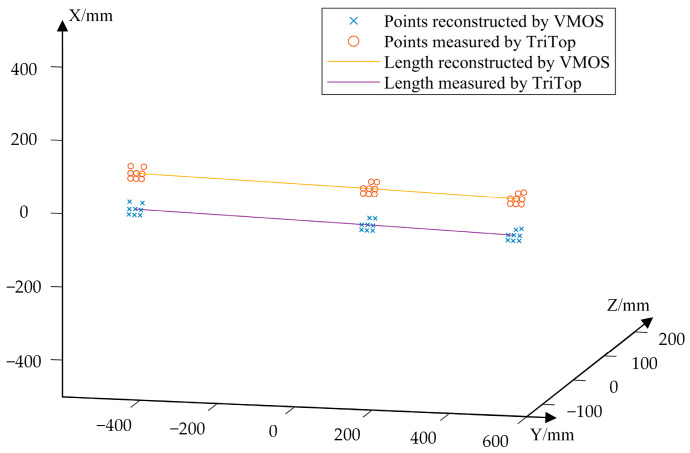
The points on the workpiece reconstructed by the VMOS and measured by TriTop^®^.

**Figure 15 sensors-23-03499-f015:**
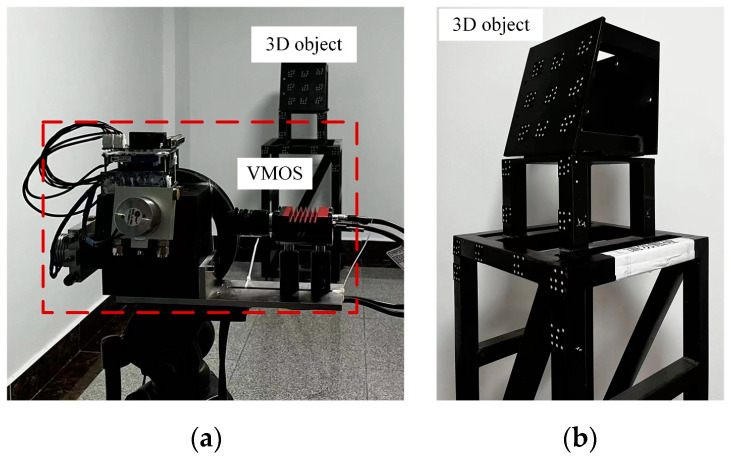
Scene of the 3D object reconstruction experiment. (**a**) The experiment scene. (**b**) The 3D object with marker points.

**Figure 16 sensors-23-03499-f016:**
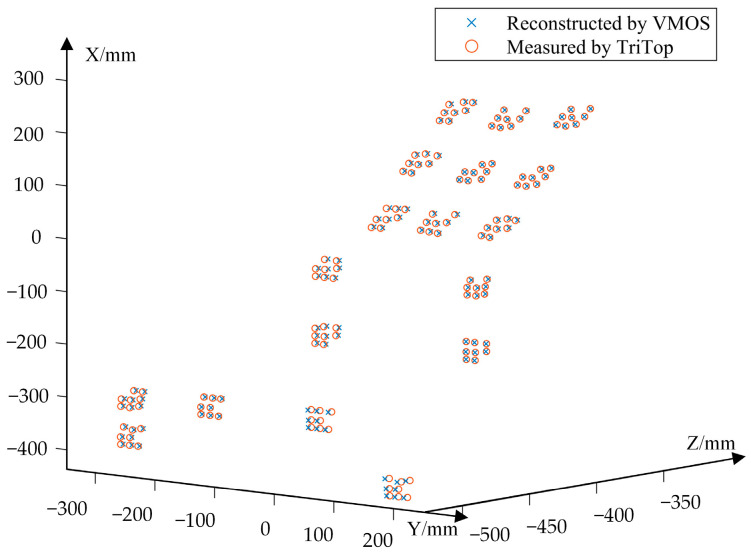
The points on the 3D structure reconstructed by the VMOS by TriTop^®^.

**Figure 17 sensors-23-03499-f017:**
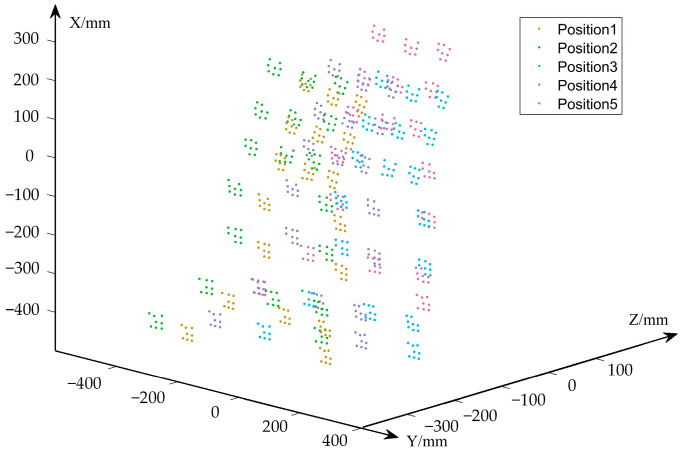
Reconstruction results of the repeatability verification experiment.

**Figure 18 sensors-23-03499-f018:**
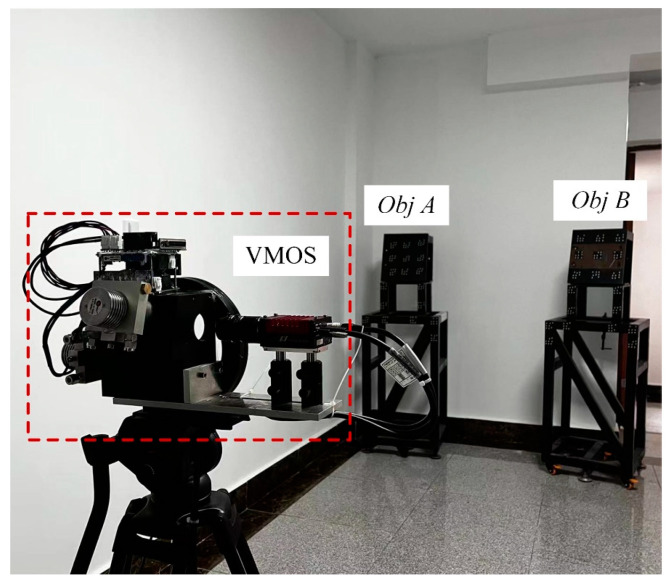
Scene of the pose estimation experiment using the VMOS.

**Figure 19 sensors-23-03499-f019:**
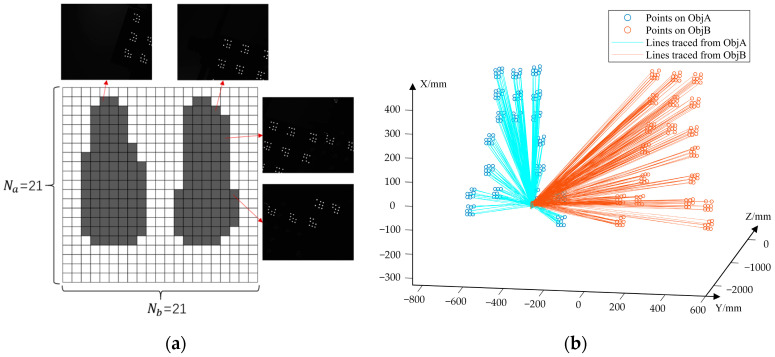
Pose estimation using the VMOS. (**a**) The virtual cameras participating in the pose estimation and the photos taken by the virtual cameras. (**b**) The lines and object point correspondences.

**Figure 20 sensors-23-03499-f020:**
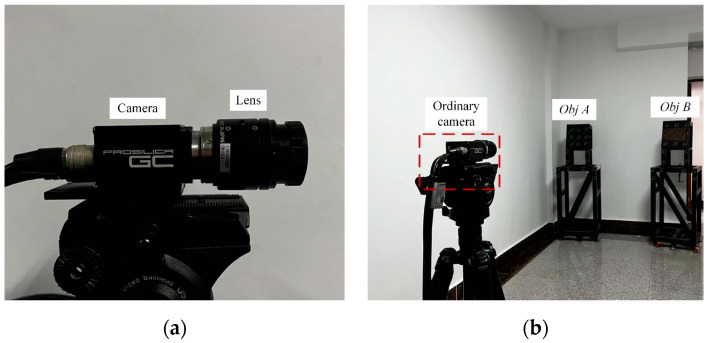
Pose estimation experiment using a single ordinary camera. (**a**) The ordinary camera system hardware setup in the experiment. (**b**) Scene of the pose estimation experiment using a single camera.

**Table 1 sensors-23-03499-t001:** The pixel errors of coded points captured by the VMOS at the same galvanometer deflection.

	MA (Pixel)	RMS (Pixel)
Horizontal	0.041	0.034
Vertical	0.043	0.035
Overall	0.067	0.038

**Table 2 sensors-23-03499-t002:** The reprojection error under the initial parameters and optimized parameters.

Direction	Initial		Optimized	
	MA (Pixel)	RMS (Pixel)	MA (Pixel)	RMS (Pixel)
Horizontal	0.206	0.143	0.200	0.132
Vertical	0.176	0.129	0.169	0.110
Overall	0.296	0.152	0.282	0.137

**Table 3 sensors-23-03499-t003:** The distance errors of the reconstructed scale bar.

	AM (mm)	RMS (mm)
Distance errors	1.007	0.835

**Table 4 sensors-23-03499-t004:** Reconstruction errors of 19*8 coded points on the 3D object.

Direction	MA (mm)	RMS (mm)
x	0.153	0.142
y	0.181	0.088
z	1.370	0.776
Full	1.404	0.769

**Table 5 sensors-23-03499-t005:** Distance errors in each pair of aligned sets of points.

	AM (mm)	RMS (mm)
Distance errors	0.913	0.616

**Table 6 sensors-23-03499-t006:** Comparison of pose estimation errors between the VMOS and the ordinary camera.

6D Pose Parameters		VMOS	Ordinary Camera
Rotation error (°)	x-axis	0.016	0.250
	y-axis	0.046	0.020
	z-axis	0.024	0.126
	Overall	0.055	0.295
Translation error (mm)	x-direction	0.376	0.098
	y-direction	0.081	1.400
	z-direction	0.597	3.249
	Overall	0.710	3.539

## Data Availability

Not applicable.

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
