# Peer review of "A Virtual Multi-Ocular 3D Reconstruction System Using a Galvanometer Scanner and a Camera"

_sensors, 2023, doi:10.3390/s23073499_

Round 1
Reviewer 1 Report
Article “A Virtual Multi‐ocular 3D Reconstruction System Using a Galvanometer Scanner and a Camera” is well written and presented very good. The proposed method is tested on different applications and made comparison with the existing technique.
Please look into the following points:
Line no. 356 and Table 1 – why there is * symbol in the superscript of direction. Please define.
In Table 2 – define the * symbol in the superscript of direction. Please define.
Author Response
Thank you for your acknowledgment of this research. We apologize for the writing problems in this paper and thank you for your careful review.
The ‘*’ symbols in Table 1 and Table 2 are not necessary and have been deleted.
Reviewer 2 Report
1. No equations 14 and 16.
2. Table 4. The rotation error of y-axis of the VMOS is worse than Ordinary Camera, the translation error of x-direction of the VMOS is worse than Ordinary Camera. Please explain. And the results do not support your description in lines 459-464.
Author Response
We apologize for the writing problems in this paper and thank you for your careful review. Based on your comments and suggestions, we have made the following changes to the manuscript. On some issues, we would like to provide some explanations.
- Due to the adjustment of the formula sequence during the revision of the manuscript, the formula number was not automatically updated, resulting in numbering errors. The formula numbers have now been revised. Proofreading on English language has been made for the revised version.
2. The overall rotation and translation errors, which are more suitable for performance comparison, are added to Table 5. The description in original lines 459-464 has been revised accordingly. The VMOS has a slightly larger transformation error in the x-direction and y-direction, the reason of which probably lies in that the accuracy in the component directions has been sacrificed to obtain a better global accuracy during the nonlinear optimization.
Reviewer 3 Report
The paper presents an interesting development of the 3D reconstruction methods with appropriate modelling, system design and well considered experiments. It will be helpful if some discussions on comparing major commercially available and mature system such as Autodesk recap pro.
Some assumptions could have been made such as the impact of the surroundings. In general, shadows, different lighting conditions, surface conditions of the scanned objects can have different impacts on modelling. The mentioned experiment has been designed and completed in a very ideal indoor setting with very limited consideration of those factors. It will be worth mentioning it in the model or at least the future/further work.
Author Response
Thank you for your recognition of our study.
In response to the first suggestion, we would like to clarify that all the 3D reconstruction experiment results of our method in the paper have been compared with TriTop, a well-established commercial 3D measurement device.
The focus of this paper is to propose a new 3D reconstruction device and to provide a simple and effective calibration method. The system is mainly for industrial inspection applications at present, which usually concern indoor settings and cooperative visual targets. According to your second suggestion, we point out in the last section that the VMOS has the potential to be used in non-laboratory environments. Application scenarios in more challenging surroundings can be explored in the future.
Reviewer 4 Report
The paper presents an approach to wide-scene stereo scene scanning based on using a pair of mirrors attached to galvanometers to steer the camera field of view, effectively producing multiple overlapping fields of view from spatially separated cameras. Largely standard geometric stereo methods are used to compute the 3D reconstructions given the virtual overlapping cameras.
The strength of the paper is the novelty of using the galvanometers for stereo, which might be useful for compact wide-field of view 3D scanning. Another strength is the virtual camera approach means that multiple views (e.g. 4) of the same target point are observed, rather than 2 in the case of normal binocular stereo.
There are some weaknesses:
1) As the galvanometers are moving parts, there is likely to be errors arising from the repeatability of obtaining the same mirror positions after each movement, which means that the calibration of each camera position may be slightly inaccurate. The paper really needs some experimental evaluation of repeatability accuracy.
2) It looks like all experiments used the calibration objects for estimating accuracy. By their design, this means that feature point correspondence is not likely to have errors. The experimental evaluation should include some reconstruction of objects other than the calibration objects, ie. where correspondence errors are present and thus might cause the proposed approach to have difficulty.
3) It appears that there are some previous similar approaches [10,35] that use a similar approach. The paper states that the proposed approach has higher accuracy, but this claim needs to be justified by some experiment. A potential cause of the difference in accuracy is a difference in camera resolutions.
4) There are some slightly similar approaches with using a galvanometer, although with a time-of-flight sensor rather than a geometric stereo sensor. These should probably be included in the literature review:
a) Hegna, T., Pettersson, H., Grujic, K., Inexpensive 3-d Laser Scanner System Based On A Galvanometer Scan Head, CloseRange10(xx-yy).
b) 3D laser scanner system based on a galvanometer scan head for high temperature applications. T Hegna, H Pettersson, KM Laundal, K Grujic - Proc. of SPIE Vol - spiedigitallibrary.org
c) Pathway to a compact, fast, and low-cost LiDAR. S Shi, L Wang, M Johnston, AU Rahman… - 2018 4th …, 2018 - ieeexplore.ieee.org
5) Line 456 should say "The errors in the parameters" rather than "The parameters".
Author Response
Thank you for your recognition and affirmation of our work. Your comments and suggestions are important to help us improve the manuscript's quality. We have made some changes to the manuscript with reference to your comments and suggestions. The followings are the details.
- Yes, the high repeatability of the galvanometer is the basis of the proposed VMOS calibration method. The high repetition accuracy is one of the excellent characteristics of the galvanometer. We have tested the repeatability of the chosen galvanometer before setting up the VMOS experimental system, and the tested repeatability meets its nominal value (<8μrad). In addition, we add an experiment in Section 3.3.3 of the revised manuscript to verify the 3D reconstruction repeatability of the VMOS.
- Our system is mainly oriented to the industrial inspection field at present, which usually concerns indoor settings and cooperative visual targets. Therefore our experiments are mainly designed for these application requirements. According to your comment, we point out in the revised last section that the VMOS has the potential to be used in non-laboratory environments. Application scenarios in more challenging surroundings can be explored in the future.
- We mention in the paper that our system outperforms the other system mentioned in the literature [10, 37] in terms of 3D reconstruction accuracy. We do not have conditions to reproduce the experiment in the literature. Instead, our judgment is based on the experimental results of the VMOS and the reported results in the literature. Yes, except for the other advantages of the VMOS, a potential cause for the higher accuracy may lie in the different image resolution. However, it should be noted that the resolution of the camera itself in the VMOS is not high. But the VMOS does take images with much detail, since each imaging region is much smaller than that of the ordinary camera. This is exactly one of the advantages of the VMOS.
- Thank you for recommending these papers to us. We have studied them, summarized the work, and added them to the introduction section.
- Thank you for your careful review. Some narrative errors in the text have been corrected.
Round 2
Reviewer 4 Report
Second review. The authors have addressed well 1 of the issues raised in the first review. However, the other 3 issues are not well addressed:
> 1) As the galvanometers are moving parts, there is likely to be errors arising from the repeatability of obtaining the same mirror positions after each movement, which means that the calibration of each camera position may be slightly inaccurate. The paper really needs some experimental evaluation of repeatability accuracy.
An additional experiment was added, where the target objects are moved and there is an estimate of the variance in the reconstruction accuracy. Unfortunately, this experiment does not address the criticism. The core issue is the repeatability of the galvanometer repositioning, under the assumption that the calibration is still accurate. What is needed is, for a single calibration, multiple observations of the same targets, after the camera view has moved to a random other position, and then returned to the original view positions. The important measurement is the variance in the computed point positions. If the sensor is to be used for industrial inspection, the repeatability of the measurements is crucial. This experiment should be easy to perform.
> 2) It looks like all experiments used the calibration objects for estimating accuracy. By their design, this means that feature point correspondence is not likely to have errors. The experimental evaluation should include some reconstruction of objects other than the calibration objects, ie. where correspondence errors are present and thus might cause the proposed approach to have difficulty.
The experiments seem to continue to use the calibration objects and point locations, which are designed to be easy to locate and correspond. Real objects have more confusing textures that can cause mis-correspondence and thus reconstruction errors. It is unclear whether the proposed approach would have larger or smaller error than a traditional approach, especially considering the introduction of error due to the galvanometer movement and repeatability. Some experiment is needed where an object with a known shape and natural texture is measured, and its measurements are compared to the known shape. This shape comparison should include an accuracy comparison with a standard stereo system using the same camera. This experiment should be easy to perform.
> 3) It appears that there is a previous approach [35] that uses a similar > approach. The paper states that the proposed approach has higher accuracy, but this claim needs to be justified by some experiment. A potential cause of the difference in accuracy is a difference in camera resolutions.
The authors claim that repeating the original experiments are not feasible. This is probably true, but it is hard to determine which approach would have better accuracy if using the same camera and optics. It is clear that modern accuracy has improved a lot because of larger pixel size sensors. This effect can dominate over lower quality algorithms. If the text were rewritten to discuss this issue more carefully, then that would be acceptable.
To summarize, I believe that the authors are presenting a largely original approach on an existing idea. However, it is not clear in what way this is a better idea than previous approaches. Evidence should be presented that it is better - otherwise future researchers and engineers might end up using an inferior approach.
Author Response
Dear reviewer
Thank you for your review of our manuscript again. Your comments and suggestions are important to help us improve the quality of our manuscript. We have made some changes to the manuscript with reference to your comments and suggestions. The followings are the specific responses.
- Yes, the high repeatability of the galvanometer is the basis of the proposed VMOS calibration method. The high repetition accuracy is one of the excellent characteristics of the galvanometer. We have conducted an experiment on the galvanometer scanner repeatability, and the new experiment is supplemented in Section 3.2.1. The experimental results show that the repeatability conforms with the nominal value of the galvanometer. The pixel error caused by repeatability is significantly smaller than the reprojection error. We hope the feasibility of the VMOS calibration method can be verified with this experiment.
- Yes, the feature extraction and matching are the difficulty in 3D reconstruction, and also the popular research area in the field of image processing. Not only our system, but other multi-ocular reconstruction systems also face the problem of correspondence errors. However, the purpose of this paper is to introduce the VMOS and its calibration method, and to verify the system reconstruction accuracy and pose estimation accuracy. In order to quantitatively validate the reconstruction performance of VMOS, we take the objects pasted with coded markers that can be accurately measured in order to analyze the reconstruction errors with ground truth. The experiments in the manuscript mainly face the industrial application scenarios that we are targeting to. For the above considerations, we have not significantly revised the article to address this comment.
- Due to the inability to replicate the system proposed in [10, 37] with existing devices for comparative experiments, the direct accuracy comparison between the VMOS and the system described in [10, 37] may not be very rigorous. Therefore, we have revised the relevant expressions in the new version.
Thank you again for your helpful comments and suggestions.
Round 3
Reviewer 4 Report
I am satisfied with the revisions.